# *Loranthus regularis* Ameliorates Neurodegenerative Factors in the Diabetic Rat Retina

**Mohammad Shamsul Ola** [1,*], **Ahmed Z. Alanazi** [2], **Ajmaluddin Malik** [1], **Abdul Malik** [3], **Mohammed Ahmed** [2], **Salim S. Al-Rejaie** [2] and **Abdullah S. Alhomida** [1]

1   Department of Biochemistry, College of Science, King Saud University, 2A62, Building 5, P.O. Box 2455, Riyadh 11451, Saudi Arabia; amalik@ksu.edu.sa (A.M.); alhomida@ksu.edu.sa (A.S.A.)
2   Department of Pharmacology and Toxicology, College of Pharmacy, King Saud University, P.O. Box 55760, Riyadh 1145, Saudi Arabia; azalanazi@ksu.edu.sa (A.Z.A.); mmahmed114@yahoo.com (M.A.); rejaie@ksu.edu.sa (S.S.A.-R.)
3   Department of Pharmacy, College of Pharmacy, King Saud University, P.O. Box 55760, Riyadh 1145, Saudi Arabia; amoinuddin@ksu.edu.sa
*   Correspondence: mola@ksu.edu.sa; Tel.: +966-558-013-579; Fax: +966-146-757-91

**Abstract:** Diabetic retinopathy remains a primary source of blindness with the growing pandemic of diabetes. Numerous studies have shown that early neurodegeneration caused by elevated oxidative stress may initiate microvascular damage in the diabetic retina during the last few decades. A variety of preventive and treatment strategies using phytochemicals that possess high antioxidants have shown great promise in reducing diabetes-induced neurodegeneration retinal damage. In this investigation, we employed an extract of Loranthus regularis, a traditional medicinal herb which is found to improve diabetes and associated complications in experimental studies. We orally treated STZ-induced diabetic rats with L. regularis and analyzed the neurodegenerative factors in the retina. After treatments, we used Western blotting techniques to analyze the protein content of neurotrophic factors (NGF, BDNF, TrkB), apoptotic factors (cytochrome c, Bcl-2, Bax), and phosphorylation of AKT in the diabetic retina. Additionally, we used ELISA methods to measure the contents of BDNF and the activity of Caspase-3 and biochemical procedures to determine the levels of glutathione and lipid peroxidation (TBARS). Our findings show that *L. regularis* treatments resulted in a considerable increase in neurotrophic factors and a decrease in apoptotic factors in the diabetic retina. Furthermore, in diabetic retina treated with *L. regularis*, the level of Bcl-2 protein increased, while the phosphor-AKT signaling improved. As a result, *L. regularis* may protect against diabetic-induced retinal neuronal damage by increasing neurotrophic support and reducing oxidative stress and apoptosis. Therefore, this study suggests that in diabetic retinopathy, *L. regularis* could be a potential therapy option for preventing neuronal cell death.

**Keywords:** diabetes; retina; neuroprotection; *Loranthus regularis*; oxidative stress; apoptosis

## 1. Introduction

Diabetic retinopathy (DR) remains a significant impediment to diabetes and a primary source of blindness and vision loss in adults. With an increasing worldwide prevalence of diabetes, which is predicted to be 700 million by 2045, the suffering of DR on both people and the burden on the healthcare system is projected to be extremely high [1,2]. Despite significant laboratory and clinical investigations, the primary etiology of diabetes-related vision loss and blindness remains unknown. The rising prevalence of diabetes demands new approaches to understanding the disease's pathogenesis to enhance retinopathy's early detection, prevention, and treatment. In recent decades, many functional tests and molecular data have demonstrated early neurodegeneration in the diabetic retina, which might contribute to microvascular damage, the classical hallmark of DR [3–5]. The precise cause(s) of early retinal neurons injury in diabetes are not known. Although numerous

reports established that diabetes increases apoptosis and oxidative stress, while decreasing neurotrophic factors, interestingly, all of these are critical markers of neurodegeneration in the diabetic retina [3,6–10].

Both retinal nerve growth factor (NGF) and brain-derived neurotrophic factor (BDNF) are known to aid in the preservation and endurance of vascular and neuronal cells (BDNF) [11–13]. Alternatively, diabetes-induced changes in these neurotrophic factors are connected to the pathogenesis of diabetic retinopathy [14,15]. Downregulation of neurotrophins or alterations in their signaling (BDNF-TrkB-Akt) could significantly change retinal and brain function [16,17]. Previously, several researchers reported lower levels of NGF and BDNF in diabetic rodent retinas [18–20]. Another central process that leads to retinal cell death in diabetes is apoptosis, which is mainly increased due to high oxidative stress. Increased oxidative stress as in the case of diabetes disrupts the mitochondrial membrane, causing an imbalance of apoptotic molecules. As a result, cytosolic pro-apoptotic Bax is translocated to the mitochondria to release cytochrome c, further escalating the apoptotic process. Anti-apoptotic Bcl-2 proteins, which regulate this process, are found to be altered in diabetic retinas [21]. Thus, targeting apoptosis as a preventive or therapeutic strategy would be an ideal approach. Several studies reported treatments to diabetic rodents with flavonoids including rutin-, hespertin- and quercetin-suppressed apoptosis and improved apoptotic regulatory proteins in the retina [8,19,22,23].

*L. regularis* is a traditional herbal plant from the Loranthaceae family primarily found in Yemen and Saudi Arabia. Mistletoes are members of the Loranthaceae family which are often used to treat hypertension, diabetes, and inflammatory disorders [24–26]. Previous studies reported that the active components of *L. regularis* served as antioxidant, antimicrobial, and anti-inflammatory factors [27,28]. More recently, our research team has investigated the potential beneficial effects of *L. regularis* against inflammation and oxidative stress in diabetes-induced hepatocellular damage and nephropathy [29,30]. *L. regularis* and its constituents, being bioactive with a potentially beneficial role in combating diabetes and its complications, here we used oral treatments of this plant extract to protect neurodegenerative factors in the retina of STZ-induced rats.

This study aimed to employ extracts of *Loranthus regularis* (*L. regularis*) to ameliorate altered neurotrophic factors, apoptotic factors, and oxidative stress in diabetic retinas, which might ameliorate neurodegeneration. Interestingly, our study suggests the beneficial effects of oral treatments of *L. regularis* to diabetic rat retinas by ameliorating diminished neurotrophic effects and reducing oxidative stress, and apoptosis, which may protect retinal neurodegeneration early in diabetes.

## 2. Materials and Methods

### 2.1. Animals

We obtained male albino Wistar rats (n = 28; weight, 250–270 g) from the Experimental Animal Care Centre, Pharmacy College, University of King Saud, Riyadh. Rats were housed at 22.1 °C with 50–55 percent humidity, and a 12 h light/dark cycle, having free access to food and water. During our animal experiments, we followed the guidelines of the National Institute of Health. The Research Ethics Committee of King Saud University approved the study (SE-19-146).

### 2.2. Induction of Diabetes

Streptozotocin (STZ) (65 mg/kg; Sigma-Aldrich, St. Louis, MO, USA) was dissolved in freshly made 0.1 M citrate buffer (pH 4.5) and a single dose was injected into 16 rats to induce diabetes. In control rats, an equivalent amount of the buffer was administered. Diabetes was validated by measuring fasting blood glucose levels with an Accu-Chek Compact-Plus glucose meter (Roche Diagnostics, Meylan, France) after two days of STZ-injection. Diabetic animals were included in the study if their blood sugar level was more than 13.9 mmol/L.

### 2.3. Preparation of L. regularis Extract

*L. regularis* leaves, flowers, and twigs were taken from the Al-Mahuit governate (Yemen). These were identified at Sana'a University's Faculty of Pharmacy's Pharmacognosy Department with voucher specimens (Mo-M05). Grinding one kilogram of air-dried plant material and extracting it for five hours with three liters of methanol using a Soxhlet device yielded a crude methanol extract of *L. regularis*. Filtering and evaporating the extract with a rotary evaporator yielded 114 g of crude methanol extract. For *L. regularis*, we used a complete extraction, fractionation, and isolation process. Three biologically active flavonoids, quercetin 3-Oa-L-rhamnopyranoside, quercetin 3-Ob-L-arabinopyranopyranopyranoside, and quercetin 3-Ob-D-galactopyranoside were separated and purified as a result of these processes [28].

### 2.4. Experimental Design

For drug treatment studies, four groups of rats were made: control (vehicle), diabetes (STZ), diabetic plus *L. regularis* (150 mg/kg/day; LRL + STZ), and diabetic plus *L. regularis* (300 mg/kg/day; LRH + STZ) (n = 6 each). One week after STZ injection, *L. regularis* extract was suspended in 0.25 percent carboxymethyl cellulose sodium (CMC) solution and administered the medication by gavage once a day for four weeks. Control and STZ-treated animals were given the same CMC solution as a vehicle. Ketamine (92 mg/kg) and xylazine (10 mg/kg) were administered intraperitoneally to anesthetize the animals and the blood samples obtained from heart puncture. The serum was separated after centrifugation at $2500 \times g$ for 15 min and stored at $-20\,^\circ$C until use.

### 2.5. Retina Isolation

After the oral treatments of *L. regularis*, the animals fasted overnight, and blood samples were taken under anesthesia through the heart. Retinas were dissected immediately and rinsed in ice-cold phosphate buffer saline to eliminate any blood or serum. Then, retinas were placed in Eppendorf tubes, flash-frozen in liquid nitrogen, and kept at $-70\,^\circ$C until the assay.

### 2.6. Determination of Retinal BDNF by ELISA

The retina was homogenized by sonication in 10 mM HEPES lysis buffer, pH 7.4, containing 0.2% SDS, 1% Triton X-100, 100 mM NaCl, and protease inhibitors.

The protein levels in retinal lysates were determined using the Bio-Rad protein assay kit (Bio-Rad, Hercules, CA, USA) after centrifugation for 10 min at $15,000 \times g$. Then, BDNF was quantified using ELISA kits (Quantikine, R&D Systems, Minneapolis, MN, USA) using equal quantities of retinal samples from LR-treated and untreated rats, based on the manufacturer's instructions.

### 2.7. Western Blotting

The retinas from all the groups were homogenized ultrasonically in 10 mM Hepes lysis buffer containing 10 mM sodium pyrophosphate, 10 mM NaF, 2 mM EDTA, 100 mM NaCl, 1 mM PMSF, 1 mM Na3VO4, and 1 mM benzamidine to determine BDNF protein level. Retinal lysates were centrifuged and supernatants were collected, and protein levels were measured according to our published methods. Protein loading samples for SDS-PAGE were made in Laemmli's sample buffer, and 25–50 μg proteins were separated on 10–12% SDS–PAGE gels and then transferred to nitrocellulose membranes. The membranes were then blocked with 5% non-fat milk prepared in Tris-buffered saline with 0.1% Tween-20 (TBS-T), rinsed, and incubated overnight with BDNF, TrkB antibodies (1 μg/mL, R&D system, Minneapolis, MN, USA); cytochrome c, and NGF antibodies (1 μg/mL; Abcam, Cambridge, MA, USA); Bax, Bcl-2 and anti-phospho-Akt (ser473) antibodies (1 μg/mL; Santa Cruz Biotechnology, Santa Cruz, CA, USA). Membranes were rinsed three times in TBS-T for five minutes each time before being incubated with respective secondary horseradish peroxidase-conjugated antibodies (1:2000, Santa Cruz Biotechnology, Inc.,

Santa Cruz, CA, USA). Membranes were then washed 4–5 times with TBS-T for five minutes each time, and the bands were seen on an LI-COR C-digit blot scanner (Biosciences, Lincoln, NV, USA) utilizing enhanced chemiluminescence (Western blotting luminol reagents). A mouse monoclonal ß-actin antibody (1:2000, Santa Cruz Biotechnology, Inc., Santa Cruz, CA, USA) was used as an internal control.

### 2.8. Glutathione (GSH) and TBAR Assay

GSH and TBARS levels were determined in the retinas of *L. regularis*-treated and untreated control and diabetic rats (Cayman Chemical Company, Ann Arbor, MI, USA) based on the supplier's protocol. The retina lysate was prepared and centrifuged at 10,000 rpm for 10 min to obtain the supernatant. TBARs and GSH were measured using a 50 μL retinal supernatant. GSH and TBARs standard curves were made, and the unknown TBARs and GSH concentrations in the samples were determined using a linear regression algorithm.

### 2.9. Caspase-3 Assay

The retinal caspase-3 activity was evaluated utilizing a caspase-3 assay kit (R&D Systems, Minneapolis, MN, USA) based on the supplier's protocol. In a 96-well microplate, the reaction for caspase activity was employed with 50 μL of rat retinal samples. The chromophore pNA is released when caspase-3 colorimetric substrate (DEVDpNA) is cleaved by the caspase and measured with a spectrophotometer at 405 nm using a microplate reader (Auto Bio Labtech Instruments, Co, Ltd., Zhengzhou, China). The results are presented in terms of an increase in optical density that correlated with an increase in caspase activity.

### 2.10. Statistical Analysis

The data of the results are evaluated as the means ± standard error of the mean (SEM). A one-way ANOVA followed by the Newman–Keuls was used for multiple comparison tests (n = 6). The Statistical Package for the Social Sciences Version 12 (SPSS 12.0) was used to conduct statistical analyses. *p*-values of less than 0.05 were considered significant.

## 3. Results

### 3.1. Effects of L. regularis on Retinal BDNF Levels

The levels of BDNF in the retinal of *L. regularis*-treated and untreated control and diabetic rats were determined by the ELISA method. In diabetic retina, the BDNF level considerably decreased compared to in non-diabetic control rats ($20.3 \pm 2.2$ vs. $11.4 \pm 1.5$ pg/mg protein; $p < 0.01$). Although, when diabetic rats were given a high dose (300 mg/kg/day) of *L. regularis* extract, the amount of BDNF in their retinas significantly increased compared to non-treated diabetic rats ($11.4 \pm 1.5$ vs. $17.4 \pm 2.1$ pg/mg protein; $p < 0.05$), (Figure 1).

### 3.2. Effects of L. regularis on Protein Expression of BDNF, proBDNF, and NGF in the Retina of Diabetic Rat

Immunoblotting techniques were used to evaluate the protein contents of BDNF, proBDNF, and NGF in the retinas of *L. regularis*-treated and untreated diabetic rats (Figure 2). Analyses of protein band using densitometry revealed a significant decrease in retinal BDNF expression level in diabetic conditions as opposed to controls ($100 \pm 8.3$ vs. $30.3 \pm 5.2\%$; $p < 0.01$) (Figure 2). Although, *L. regularis* treatments resulted in a considerable rise in BDNF levels in the diabetic retina ($30.3 \pm 5.2$ vs. $60.4 \pm 7.4\%$; $p < 0.05$). Likewise, the amount of NGF in diabetic retinas was considerably lower than that of control retinas ($100 \pm 7.0$ vs. $55 \pm 5.3\%$; $p < 0.01$), while the *L. regularis* administration enhanced the level of NGF in the diabetic retina ($55 \pm 5.3\%$ vs. $75 \pm 7.2\%$; $p < 0.05$). As compared to non-diabetic rats, proBDNF level was substantially higher in the diabetes retina ($100 \pm 6.4$ vs. $250 \pm 18.5\%$; $p < 0.01$) (Figure 2). Conversely, the amount of proBDNF was markedly reduced in the *L. regularis*-treated diabetic retinas than without treatments ($250 \pm 18.5$ vs. $200 \pm 15.2\%$; $p < 0.05$).

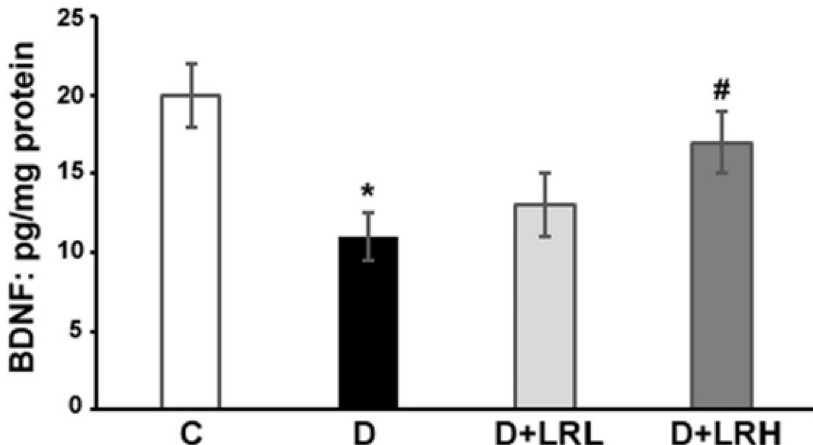

**Figure 1.** BDNF levels in the retinas of *L. regularis*-treated diabetic and control rats. Retinal BDNF levels were measured using an ELISA kit. BDNF levels in diabetic retinas were compared with *L. regularis*-treated diabetic rats and control rats. The values are means ± SEM (standard error of mean) (* $p < 0.01$ vs. control; # $p < 0.05$ vs. diabetes, N = 6). C (control), D (diabetic) D + LRL (low-dose *L. regularis*-treated diabetic rats), and D + LRH (high-dose *L. regularis*-treated diabetic rats. The experiment was repeated twice.

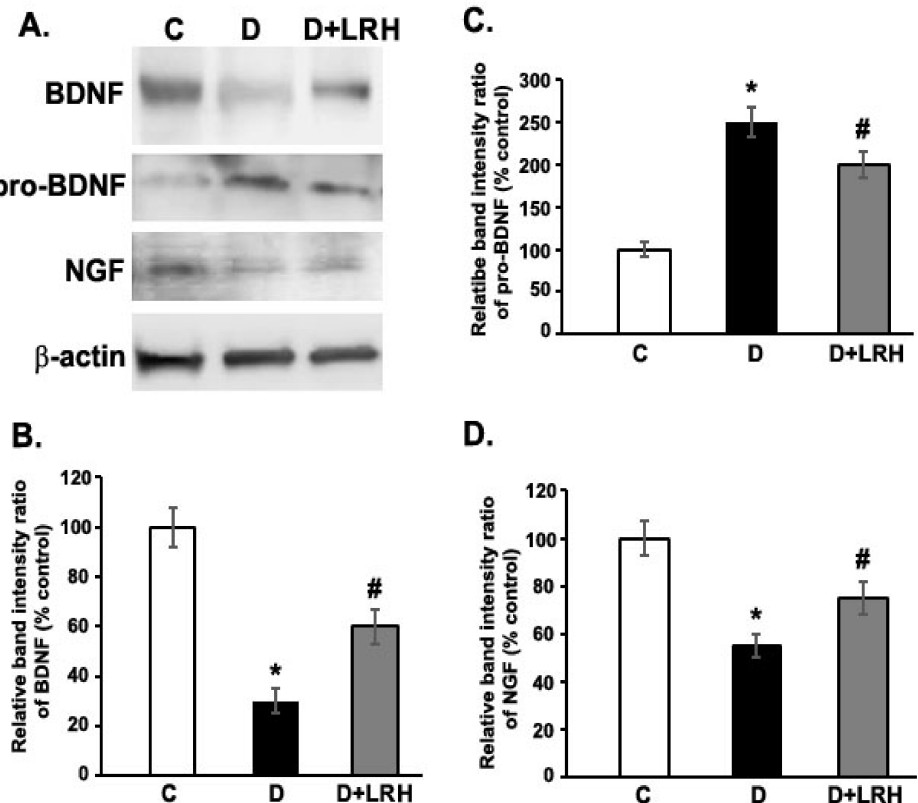

**Figure 2.** Expression of BDNF, pro-BDNF, and NGF in the high-dose *L. regularis* (LRH)-treated diabetic rats' retinas. Densitometry was used to measure the band intensities. Representative bands of the blots of BDNF, pro-BDNF, NGF, and β-actin are shown in Panel (**A**). Band intensity ratios of BDNF protein bands (Panel (**B**)), Pro-BDNF protein bands (Panel (**C**)), and NGF protein bands are displayed as a percentage of control (Panel (**D**)). The values are means ± SEM for six determinations. * $p < 0.01$, suggests a significant difference from their controls; # $p < 0.05$ suggests a significant difference from diabetic retinas. Experiments with immunoblotting were conducted twice.

### 3.3. L. regularis Upregulates TrkB and Phospho-Akt Protein Expression in the Diabetic Retina

When we compared the protein expression of TrkB in the diabetic retina to the control retina by Western blot, the level was significantly lower ($100 \pm 9.4$ vs. $30 \pm 3.2\%$; $p < 0.01$). Moreover, in the *L. regularis*-administered diabetic groups, the expression level was significantly elevated (30 to 50%) (Figure 3). The stimulation of the Akt survival pathway by *L. regularis* reflected the improvement of the BDNF signaling pathway. The diabetic retina's Akt protein phosphorylation was reduced by over 25%, but *L. regularis* increased it to nearly normal levels (Figure 3).

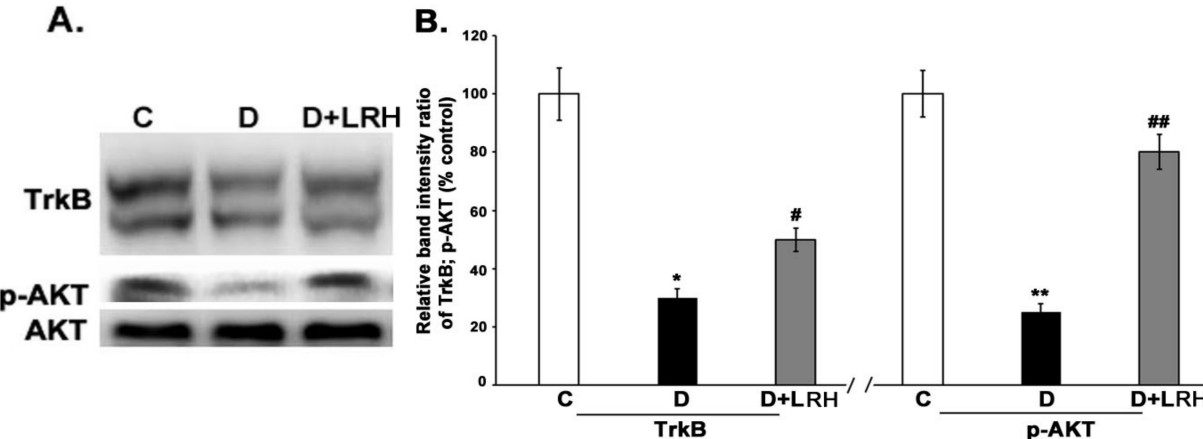

**Figure 3.** Retinal protein expression of TrkB, AKT, and p-AKT in *L. regularis*-treated diabetic rats. Densitometry was used to measure the band intensities. Representative immunoblots of TrkB, AKT, p-AKT are shown in Panel (**A**). The bar graph is provided as a percentage of control of TrkB and p-AKT protein band intensities ratios (Panel (**B**)). Values are means $\pm$ SEM for six determinations. *,** $p < 0.01$ shows a significant variation from their controls; #,## $p < 0.05$ suggests a marked difference from diabetic retinas.

### 3.4. Effects of L. regularis Treatments on Retinal Bcl-2, Bax, and Cytochrome C Protein Expression in Diabetic Rats

The expression of pro-apoptotic (Bax and Cytochrome C) and anti-apoptotic Bcl-2 proteins in the retinas of control, diabetic, and *L. regularis*-treated diabetic rats were investigated using Western blot analysis (Figure 4). The retinal level of Bcl-2 protein in the diabetic rats was considerably lower than in controls, based on densitometric examination of the bands ($100 \pm 9.2$ vs. $29.6 \pm 3.9\%$; $p < 0.01$). However, when diabetic rats were treated with *L. regularis*, the lowered level of Bcl-2 was dramatically increased ($29.6 \pm 3.9$ vs. $59.2 \pm 5.2\%$; $p < 0.05$). Furthermore, diabetic retinas had considerably higher levels of pro-apoptotic Bax and cytochrome c expression than controls ($p < 0.01$). However, treating diabetic rats with *L. regularis* reduced the retinal Bax and cytochrome c expression almost to control levels ($p < 0.05$). (Figure 4).

### 3.5. Effect of L. regularis on Glutathione and TBARs Level

Glutathione (GSH) and TBARS are well-known cellular markers of oxidative stress.

The levels of antioxidant GSH, and the lipid peroxidation indicator TBARs, were measured in the retinas of *L. regularis*-treated and untreated diabetic rats. In diabetic retinas, the amount of GSH was considerably lower ($p < 0.05$) than in controls. However, only a high dose of *L. regularis* treatments significantly enhanced retinal GSH levels in diabetic rats when compared to untreated diabetic rats ($p < 0.05$) (Figure 5A). TBARs were considerably ($p < 0.01$) elevated in the diabetic retinas, while *L. regularis* high-dose (300 mg/kg/day) treatments to diabetic rats ameliorated the increased level of TBARs ($p < 0.01$, Figure 5B).

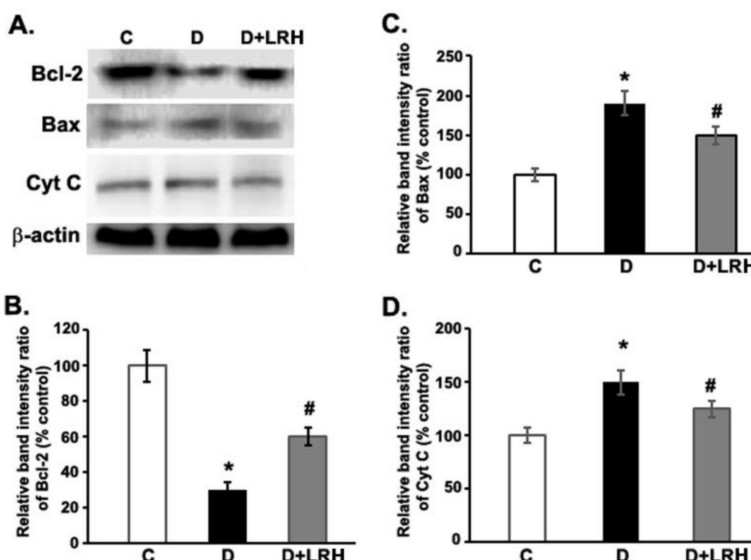

**Figure 4.** *L. regularis* ameliorated anti-apoptotic and pro-apoptotic proteins expression in the diabetic rat retina. Densitometry was used to measure the band intensities after Western blotting. Bcl-2, Bax, Cyt C, and β-actin immunoblots are shown in Panel (**A**). Band intensity ratios of Bcl-2 protein bands (Panel (**B**)), Bax protein bands (Panel (**C**)), and Cyt C protein bands displayed as a percentage of control (Panel (**D**)). The values are means ± SEM for six determinations. * $p < 0.01$ shows a remarkable variation from their controls; # $p < 0.05$ shows significantly different from diabetic retinas.

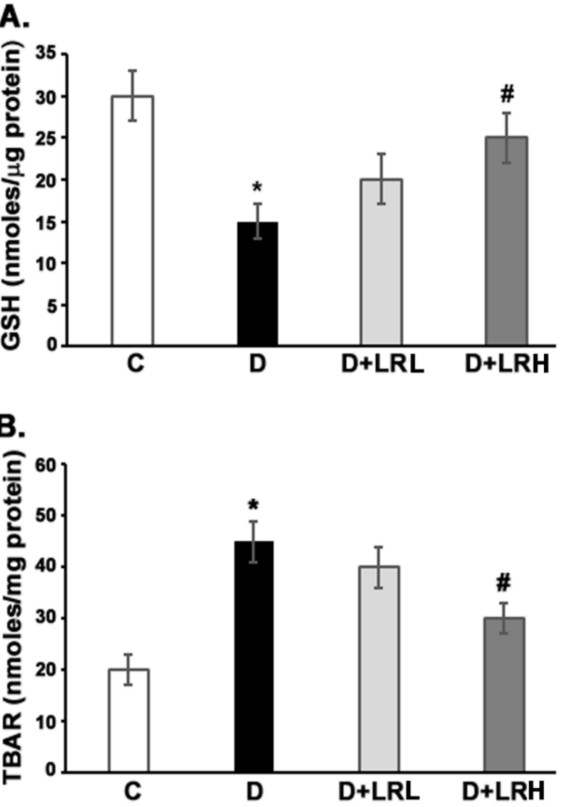

**Figure 5.** Antioxidant effects of *L. regularis* in the retinas of diabetic rats. GSH levels in all treated and untreated groups were compared as shown in panel (**A**). Similarly, the levels of TBARS in the *L. regularis*-treated and untreated groups were compared as shown in panel (**B**). The values are means ± SEM (n = 6). (* $p < 0.01$, vs. diabetic; # $p < 0.05$, vs. controls). (C) stands for control, (D) for diabetic, (LRL) low-dosage *L. regularis*, and (LRH) high-dose *L. regularis*. Experiments were carried out twice.

### 3.6. Caspase-3 Activity

The activity of the apoptotic caspase-3 enzyme in diabetic rats' retinas was measured and compared to that of control and diabetic rats treated with *L. regularis*. The activity of the caspase-3 enzyme was considerably higher in diabetic rats' retinas as compared to controls ($p < 0.01$) (Figure 6). However, *L. regularis* administration of diabetic rats reduced retinal caspase-3 activity to near-control levels ($p < 0.05$).

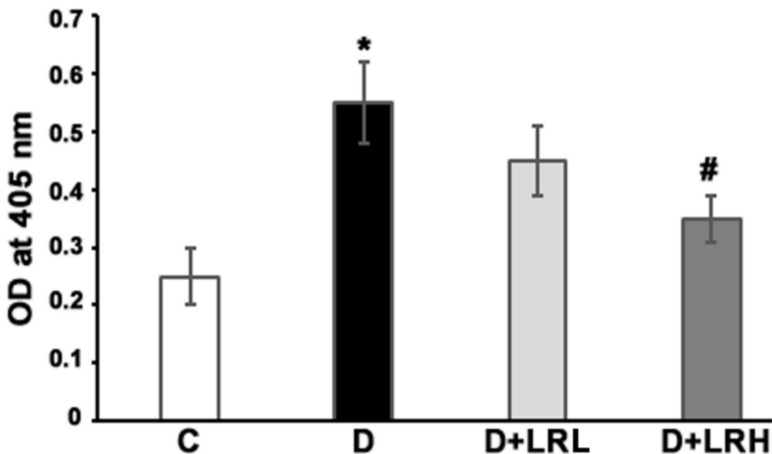

**Figure 6.** *L. regularis* inhibits caspase-3 activity in diabetic rat retinas. The activity of caspase-3 was evaluated by colorimetric enzymatic assay in the diabetic rats treated with *L. regularis*. The results are represented as a fold increase in caspase activity as assessed by an increase in optical density. (* $p < 0.01$ vs. diabetes; # $p < 0.05$ vs. control). The values are the means ± SEM (n = 6). Experiments were carried out twice. (LRL, *L. regularis* low dose; LRH, *L. regularis* high dose).

### 4. Discussion

*L. regularis* is a medicinal plant that has been used to cure a variety of ailments, including diabetes and inflammatory conditions [24,27,31]. Previously, our group investigated the beneficial effects of *L. regularis* in diabetic animals towards reducing blood glucose levels, oxidative stress, and inflammatory cytokines indicators. Other species of the *L. regularis* family (Loranthaceae) have also shown comparable antidiabetic, antioxidant, and anti-inflammatory properties [24,31]. Owing to its beneficial effects, in this study, we investigated the protective outcome of *L. regularis* extract in the streptozotocin-administered diabetic rat retinas. We determined that *L. regularis* extract treatments improved the altered levels of neurotrophic factors and signaling in diabetic retinas. Furthermore, *L. regularis* treatments reduced pro-apoptotic Bax, Cytochrome C, and caspase-3 proteins, and increased the anti-apoptotic Bcl-2 level in the retina of diabetic rats. This treatment also improved the antioxidant status by increasing GSH levels and decreasing lipid peroxidation in the diabetic retina.

Neurotrophic factors are vital for the growth and preservation of neurons. Diminished expression levels of NGF and BDNF were demonstrated to influence systemically, compromising insulin action and altering lipid and glucose metabolism, while causing tissue-specific damages, particularly in diabetic neural retinas [20,32]. Previous studies from our laboratory, including with others, have shown considerable BDNF levels decline in diabetic rat retinas and also humans in comparison to controls [18,33]. Consistent with previous works, we analyzed a lower amount of retinal NGF and BDNF in diabetic rats. Notably, administration of *L. regularis* to diabetic rats remarkably induced their expression levels in the retinas. These results agree with a few other reports stating that polyphenolic compounds may regulate the neurotrophic factors in the brain and retinas [19,34]. Furthermore, there was a reduced level of TrkB, a BDNF-specific receptor, which might aggravate glutamate excitotoxicity to retinal neurons in diabetes [35]. Notably, *L. regularis* treatments increased both TrkB expression and AKT phosphorylation. The phosphorylation of Akt is

found to reduce oxidative stress and apoptotic signals while increasing cell survival signals [36]. The anti-apoptotic effects of *L. regularis* could be attributed to diabetic rats having higher levels of retinal p-AKT. An earlier report indicated that flavonoid, epicatechin, protected the diabetic neuro-retina by enhancing the level of NGF and its signaling [37]. Thus, *L. regularis* may have neuroprotective benefits in diabetic retinas by enhancing neurotrophic support and signaling molecules.

Apoptosis is a well-known characteristic of early neurodegeneration in diabetic retinas. In the diabetic retina, pro-apoptotic molecules such as caspase-3, Bax, and cytochrome c activate other caspases to cause neurodegeneration, while anti-apoptotic Bcl-2 is decreased [3,21]. Indeed, the level of pro-apoptotic molecules increased, whereas anti-apoptotic Bcl-2 protein decreased, which agrees with a few earlier findings. Interestingly, the altered levels of apoptotic proteins are improved by *L. regularis*. Similar to previous studies, this herbal extract also suppressed caspases and attenuated apoptotic proteins in the retina of diabetic rats [19,22,33]. Therefore, *L. regularis* may repair neuronal damage in diabetic retinas by exerting anti-apoptotic actions via the BDNF–TrkB/p-Akt signaling pathway.

Flavonoids are recognized for their antioxidant properties, which inhibit diabetic retinal neurodegeneration [10,33,38]. The main constituents of *L. regularis* are identified as quercetin derivatives, including quercetin 3-O-b-L-galactopyranoside, quercetin 3-O-b-L-arabinopyranoside, and quercetin 3-O-a-L-rhamnopyranoside [39]. They have significant antioxidant and anti-inflammatory properties [12,39,40]. In STZ-induced diabetic rat retinas, *L. regularis* treatment increased glutathione and decreased lipid peroxidation. Thus, our findings suggest that supplementing with *L. regularis* may protect diabetic retinal neurons by reducing oxidative stress, pro-apoptotic caspases, and activating neurotrophic support.

## 5. Conclusions

Our results show that *L. regularis* extract decreased oxidative stress, apoptosis while increasing neurotrophic effects in the retinas of diabetic rats. The active ingredients of *L. regularis* extract, mainly quercetin glucosides, might be directly responsible for these positive effects. However, further investigation is required to study the protective role of *L. regularis* and its constituents in diabetes and its complications, particularly in preventing early retinal damage in the diabetic retinas.

**Author Contributions:** All authors participated in the design and interpretation of the study, the analysis of the data and the review of the manuscript. Conceptualization, M.S.O. and A.S.A.; methodology, A.M. (Ajmaluddin Malik) and A.M. (Abdul Malik); investigation, M.S.O. and A.Z.A.; resources, S.S.A.-R. and A.S.A.; data curation, M.A. and A.M. (Abdul Malik); writing—original draft preparation, M.S.O.; writing—review and editing, M.S.O. and S.S.A.-R.; supervision, M.S.O. and A.S.A.; project administration, A.Z.A.; funding acquisition, A.M. (Abdul Malik). All authors have read and agreed to the published version of the manuscript.

**Funding:** This research is funded by Research supporting project (RSP-2021/376), King Saud university, Riyadh, Saudi Arabia.

**Institutional Review Board Statement:** The Research Ethics Committee of King Saud University approved the study (SE-19-146).

**Informed Consent Statement:** Not applicable.

**Data Availability Statement:** Most part of the data is presented in the paper. However, any additional data can be obtained from the corresponding author on personal request.

**Acknowledgments:** The authors would like to thank King Saud University for funding this work through research supporting project (RSP-2021/376), Riyadh, Saudi Arabia.

**Conflicts of Interest:** The authors declare no conflict of interest.

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
