# Peer review of "Loranthus regularis Ameliorates Neurodegenerative Factors in the Diabetic Rat Retina"

_applsci, doi:10.3390/app12062875_

Round 1

Reviewer 1 Report

The revised manuscript, entitled “Loranthus regularis inhibits neurodegenerative factors in the diabetic rat retina” by Mohammad Shamsul Ola et al.  is interesting and generally well written but suffers from some flaws:

  1. Line 182: (30.3 ± 2.2 vs. 17.4 ± 2.1 pg/mg protein, p <0.05), (Fig. 1). Is it correct?
  2. line 200 (100 200 ± 6.4 vs. 250 ± 18.5; p < 0.01) Please add %
  3. L. regularis extract was administered once per day orally by gavage for four weeks. Please give reasons for choosing four weeks.
  4. The main conclusion of this study is: “Our results show that L. regularis extract reduced oxidative stress, apoptosis, and increased neurotrophic effects in the retinas of STZ-induced diabetic rats” However, the title of this study is focused on the neurodegenerative factors? I propose a change in the title e.g. ameliorates neurodegeneration in the diabetic rat retina …. or something else?
  5. The aim of this study should be at the end of the introduction section and please indicate the novelty of this study

Author Response

Reviewer-1

The revised manuscript, entitled “Loranthus regularis inhibits neurodegenerative factors in the diabetic rat retina” by Mohammad Shamsul Ola et al.  is interesting and generally well written but suffers from some flaws:

Reply: We would like to thank the reviewer for giving a good remark about the study and also for giving us an opportunity to edit the manuscript. We are indeed, thankful to the reviewer for making constructive comments and suggestions.

  1. Line 182: (30.3 ± 2.2 vs. 17.4 ± 2.1 pg/mg protein, p <0.05), (Fig. 1). Is it correct?

Reply: Our apology for the mistake. It would be 11.4±1.5 in place of 30.3 ± 2.2, it was an error; which is corrected now.

2. line 200 (100 200 ± 6.4 vs. 250 ± 18.5; p < 0.01) Please add %

Reply: Yes, % is added. Thanks

3. L. regularis extract was administered once per day orally by gavage for four weeks. Please give reasons for choosing four weeks.

Reply: Thanks for this question. Our study is designed based on previous studies by our group who have published at least two papers in the peer-reviewed journals cited in the text and also below here, where they administered the doses of L.regularis once per day orally by gavage for four weeks. They observed significant ameliorative effects in diabetes and its complications. In this study, we used similar doses and methods of treatment to analyze protective effects in the retina. Generally, early molecular changes start appearing after a week of diabetes in the neural retina and get prominent effects after 3-4 weeks of diabetes. Therefore, this time frame is selected for the treatments. References are cited in the text and below here.

Ref No: 29; Alanazi, A.Z.; Mohany, M.; Alasmari, F.; Mothana, R.A.A.; Alshehri, A.O.A.; Alhazzani, K.; Ahmed, M.M.; Al-Rejaie, S.S. Amelioration of Diabetes-Induced Nephropathy by Loranthus regularis: Implica-tion of Oxidative Stress, Inflammation, and Hyperlipidaemia. Appl. Sci. 2021, 11, 4548

Ref No: 30; Alanazi, A.Z Protective Role of Loranthus regularis against Liver Dysfunction, Inflammation, and Oxidative Stress in Streptozotocin Diabetic Rat Model Evidence-Based Complementary and Alternative Medicine Volume 2020, Article ID 5027986, https://doi.org/10.1155/2020/5027986

4. The main conclusion of this study is: “Our results show that L. regularis extract reduced oxidative stress, apoptosis, and increased neurotrophic effects in the retinas of STZ-induced diabetic rats” However, the title of this study is focused on the neurodegenerative factors? I propose a change in the title e.g. ameliorates neurodegeneration in the diabetic rat retina …. or something else?

Reply: Yes, we agree, we have amended the title.

5. The aim of this study should be at the end of the introduction section and please indicate the novelty of this study.

Reply: Thanks for the suggestion. Yes, we have added aim at the end as suggested and also indicated the novelty of the study.

Reviewer 2 Report

1. Animal model

It is difficult to replicate STZ induced diabetic rat model to achieve 100% success rate. If the author's work cannot achieve 100% success rate, there is a problem with the number of animals. If the author selected the rats meet blood glucose standard to do the experiment, please clarified it  .

2. Experimental design

(1) There is a time course in the pathogenesis of diabetic retinopathy in STZ rats. Retinal vascular injury usually occurs after 3 months or more lately. In this paper, the rats were intervened within 4 weeks after the increase of blood glucose, and then the animals were killed to extract the retina for molecular biological observation. At this period, there may be no retinopathy in STZ rat.

If the authors simultaneously observe the course of disease, severity of diabetic retinopathy with the change of neurotrophic factors, apoptotic factors and d phosphorylation of AKT, it will be more meaningful.

(2) The experimental group divided is unreasonable. This manuscript is divided into low-dose group (150mg) and high-dose group (300mg) according to the dose, which is hard to reflect the dose effect of the drug.

(3) This manuscript only observed the changes of relevant cytokines and other indicators 4 weeks after the successful replication of the model, lacking the time process.

3. Experimental results

Fig. 2, 3 and 4 only the comparison between the high-dose group and the control group, there is no result in the low-dose group, and the result is incomplete.

Author Response

Thanks to the reviewer for giving us an opportunity to edit the manuscript and also for making constructive comments and suggestions. Here are our replies.

  1. Animal model

It is difficult to replicate STZ induced diabetic rat model to achieve a 100% success rate. If the author's work cannot achieve a 100% success rate, there is a problem with the number of animals. If the author selected the rats to meet blood glucose standard to do the experiment, please clarified it.

Reply: Thanks for your remark. Yes, we 100% agree with the reviewer. All STZ-induced rats don’t become 100% diabetic. We always keep a few extra rats to make them diabetic so that we at least get the number required for experimental purposes. We reported six diabetic rats in the group. But we injected STZ to at least 8 rats to make them diabetic. 1-2 rats either don’t become diabetic or die after becoming diabetic. We have edited the number of rats in the revised text.

2. Experimental design

(1) There is a time course in the pathogenesis of diabetic retinopathy in STZ rats. The retinal vascular injury usually occurs after 3 months or more lately. In this paper, the rats were intervened within 4 weeks after the increase of blood glucose, and then the animals were killed to extract the retina for molecular biological observation. At this period, there may be no retinopathy in STZ rat.

If the authors simultaneously observe the course of disease, severity of diabetic retinopathy with the change of neurotrophic factors, apoptotic factors and d phosphorylation of AKT, it will be more meaningful.

Reply: Based on our research experience and from others who have been using the STZ-model of diabetic rats, the fact is that unfortunately, they don’t develop clinical retinopathy or even vasculopathy after 3 months of diabetes. Our major aim of this study was to analyze the protective effects of the drug on early retinal neurodegeneration. Early cellular and molecular signs of neural retinal damages have been observed as early as after one week of STZ-diabetes. This is the reason, we have induced STZ-diabetes in rats and after a week of diabetes stabilization, we started the drug treatments for a little longer period (4 weeks) to observe the ameliorative effects especially on neurodegenerative factors. Numerous previous studies also suggest not many graded changes in the neural retina in the early phases of diabetes in rodents. We could not prolong or keep those diabetic and treated rats for a longer period as mortality rates were higher in our conditions.  

(2) The experimental group divided is unreasonable. This manuscript is divided into low-dose group (150mg) and high-dose group (300mg) according to the dose, which is hard to reflect the dose effect of the drug.

Reply: This study was planned based on previous studies by our group who published at-least two papers in peer-reviewed journals where they selected the two doses (150mg and 300 mg/kg) of L.regularis to observe ameliorative effects in diabetes and its complications. In this study, we used the same doses of the drug to analyze protective effects in the retina. Based on the references below.

Alanazi, A.Z.; Mohany, M.; Alasmari, F.; Mothana, R.A.A.; Alshehri, A.O.A.; Alhazzani, K.; Ahmed, M.M.; Al-Rejaie, S.S. Amelioration of Diabetes-Induced Nephropathy by Loranthus regularis: Implica-tion of Oxidative Stress, Inflammation and Hyperlipidaemia. Appl. Sci. 2021, 11, 4548

Protective Role of Loranthus regularis against Liver Dysfunction, Inflammation, and Oxidative  Stress in Streptozotocin Diabetic Rat Model Evidence-Based Complementary and Alternative Medicine Volume 2020, Article ID 5027986, 8 pages https://doi.org/10.1155/2020/5027986

(3) This manuscript only observed the changes of relevant cytokines and other indicators 4 weeks after the successful replication of the model, lacking the time process.

Reply: As stated above, we aimed to observe neuroprotection by the drug as the neurodegeneration begins early in the diabetic retina. We did not measure cytokines, rather analyzed neuronal markers (BDNF, NGF, AKT, apoptosis, and oxidative stress) which indicates neuronal health early in the phase of diabetic retina whose altered level may cause vascular damage as clinically observed later in the disease progression. Our focus of research has always been more towards prevention compared to treatments since mostly neuronal damages are irreversible. Hence in this study, early neuroprotection was our main aim.

3. Experimental results

Fig. 2, 3, and 4 only the comparison between the high-dose group and the control group, there is no result in the low-dose group, and the result is incomplete.

Reply: We analyzed the effects of high and low doses of L.regularis in the diabetic retina. But our experimental results indicated that low doses (150mg) could not significantly alter the level of BDNF, oxidative stress (GSH and TBARs), and also apoptosis (Figs 1, 5, and 6). In contrast, a high dose (300mg) could significantly ameliorate their altered levels in the diabetic retina. Therefore, for Western blot experiments, we used only the high treatments group. In previous studies using the same drug, our group also reported that a high dose (300mg) was more effective in lowering diabetes and its complications as compared to a low dose. (Alanazi et al. 2021, 2020)